# Monitoring Myelin Lipid Composition and the Structure of Myelinated Fibers Reveals a Maturation Delay in CMT1A

**DOI:** 10.3390/ijms252011244

**Published:** 2024-10-19

**Authors:** Giovanna Capodivento, Mattia Camera, Nara Liessi, Anna Trada, Doriana Debellis, Angelo Schenone, Andrea Armirotti, Davide Visigalli, Lucilla Nobbio

**Affiliations:** 1IRCCS Ospedale Policlinico San Martino, Largo Rosanna Benzi 10, 16132 Genova, Italy; giovanna.capodivento@hsanmartino.it (G.C.); aschenone@neurologia.unige.it (A.S.); 2Department of Neurosciences, Rehabilitation, Ophthalmology, Genetics, and Maternal and Children’s Sciences (DINOGMI), University of Genoa, 16126 Genova, Italy; mattia.camera@edu.unige.it (M.C.); tra.anna511@gmail.com (A.T.); 3Analytical Chemistry Facility, Istituto Italiano di Tecnologia, Via Morego 30, 16163 Genova, Italy; nara.liessi@iit.it (N.L.); andrea.armirotti@iit.it (A.A.); 4Electron Microscopy Facility, IIT, Via Morego 30, 16163 Genova, Italy; doriana.debellis@iit.it

**Keywords:** CMT1A, myelin maturation, axonal growth, myelin lipids, nervous system development, sphingolipids

## Abstract

Findings accumulated over time show that neurophysiological, neuropathological, and molecular alterations are present in CMT1A and support the dysmyelinating rather than demyelinating nature of this neuropathy. Moreover, uniform slowing of nerve conduction velocity is already manifest in CMT1A children and does not improve throughout their life. This evidence and our previous studies displaying aberrant myelin composition and structure in adult CMT1A rats prompt us to hypothesize a myelin and axon developmental defect in the CMT1A peripheral nervous system. Peripheral myelination begins during the early stages of development in mammals and, during this process, chemical and structural features of myelinated fibers (MFs) evolve towards a mature phenotype; deficiencies within this self-modulating circuit can cause its blockage. Therefore, to shed light on pathophysiological mechanisms that occur during development, and to investigate the relationship among axonal, myelin, and lipidome deficiencies in CMT1A, we extensively analyzed the evolution of both myelin lipid profile and MF structure in WT and CMT1A rats. Lipidomic analysis revealed a delayed maturation of CMT1A myelin already detectable at P10 characterized by a deprivation of sphingolipid species such as hexosylceramides and long-chain sphingomyelins, whose concentration physiologically increases in WT, and an increase in lipids typical of unspecialized plasma membranes, including phosphatidylcholines and phosphatidylethanolamines. Consistently, advanced morphometric analysis on more than 130,000 MFs revealed a delay in the evolution of CMT1A axon and myelin geometric parameters, appearing concomitantly with lipid impairment. We here demonstrate that, during normal development, MFs undergo a continuous maturation process in both chemical composition and physical structure, but these processes are delayed in CMT1A.

## 1. Introduction

Charcot–Marie–Tooth disease type 1A (CMT1A) is the most common inherited myelin disorder of the peripheral nervous system (PNS), caused by the duplication of a large DNA segment of 1.5 Mb containing the *PMP22* gene. Although the genetic defect has been known for a long time, the pathomechanisms underlying the neurological phenotype are still an intriguing matter for discussion [1,2,3].

Dysmyelination rather than demyelination in CMT1A has been recognized in the literature, albeit developmental myelin and axonal deficiencies as critical features in the pathogenesis of the disease have not been deeply investigated yet [4,5]. CMT1A patients display a remarkable and uniform nerve conduction velocity (NCV) slowing that does not significantly change throughout their lives. The mean NCV is around 20 m/s and is similar in nerves of both upper and lower limbs and among different patients [3,5,6,7,8]; moreover, CMT1A patients almost never show temporal dispersion and conduction blocks [7]. This highly preserved electrophysiological picture is rather inconsistent with a typical demyelinating phenotype which instead occurs in acute and chronic inflammatory demyelinating polyneuropathies [5,6]. Interestingly, the uniform reduction of NCV in CMT1A is detectable starting from the first years of life and is stable for decades with signs of dysmyelination already present in children [7,8,9,10]. In this regard, CMT1A electrophysiology also challenges the theory of repetitive de/remyelination, that sometimes has been reported to cause onion bulb formations, in favor of an altered differentiation of Schwann cells (SCs) which likely remain in excess during development [5,7,11,12].

Nerve conduction properties are tightly dependent on myelinated fibers (MFs) and geometric parameters; in fact, altered CMT1A neurophysiology reflects pathological nerve abnormalities including the reduced myelin thickness of large fibers, the increased myelin thickness of small fibers, and shortened internodes [13,14]. Myelination and specific axon–glia interaction tightly regulate the increase in axonal diameter during development [15,16,17]. For this reason, axonal maturation defects might be expected to occur in CMT1A neuropathy. Indeed, CMTIA peripheral nerves display an altered distribution of axon populations, with a reduction in larger axons and an increase in the smaller ones [15,18,19,20].

Until now, axonal flaw was considered secondary to myelin abnormalities [21]. However, NCV studies and neuropathological evidence support a defect of PNS development in CMT1A likely involving both axon and myelin maturation.

In addition to electrophysiologic and neuropathological abnormalities, molecular alterations are also present in CMT1A; among them, an upregulation of markers of immature SCs and a downregulation of lipid metabolism genes have been found in CMT1A rats [18,21]. Consistently, we recently demonstrated that the lipid composition of peripheral myelin is also altered in CMT1A [22]. Of note, an impaired myelin lipidome affects nerve fiber architecture and electrophysiological properties. In fact, interfering with lipid biosynthetic pathways improves both the altered geometric parameters of CMT1A MFs and neurophysiology [21,22].

Overall, data from the literature show that neurophysiological, neuropathological, and molecular alterations are present in CMT1A, but when they arise and become relevant has not been investigated. To shed light on pathophysiological mechanisms that occur during development, and to investigate the relationship among axonal, myelin, and lipidome deficiencies in CMT1A, we extensively analyzed the evolution of both myelin lipid profile and MF structure in WT and CMT1A rats. We found that CMT1A peripheral nerves display an immature phenotype, characterized by a highly consistent maturation delay in both myelin lipidome and axon-related morphometric parameters.

## 2. Results

### 2.1. Maturation of Myelin Lipid Composition Is Delayed in CMT1A

We recently demonstrated that the lipid composition and structure of peripheral myelin are altered in adult CMT1A rats [22]. Whether this alteration is due to impaired myelin maturation or a maladaptive response to chronic nerve damage during disease progression remains to be proven. To address this issue, we first monitored the lipid profile of WT and CMT1A myelin during development from P5 to P180 using high-resolution liquid chromatography with tandem mass spectrometry (LC-MS/MS).

The PCA score plots of lipidomic data show that in both WT and transgenic rats myelin lipid composition changes over time, indicating a continuous maturation process (Figure 1a and Appendix A). While differences between genotypes are not so obvious within the first ten days after birth, beginning from P20 onwards CMT1A myelin significantly diverges from the WT one (Figure 1b,c). Consistently, when P5-P180 CMT1A lipid profiles were individually plotted on the WT ones, starting from P20, they were shifted towards earlier developmental stages (Appendix A). Interestingly, we found that P20 CMT1A myelin is characterized by a marked decrease in almost every sphingomyelin (SM) and hexosylceramide (HexCer) species compared to WT (Appendix A), together with an increase in acylcarnitines (CAR) and some glycerophospholipid (GPL) species (Appendix A). Indeed, several SM species show a decrease in CMT1A myelin at P5 and are significantly reduced at P10 (Supplementary Appendix A), indicating that myelin abnormalities, even if subtler, are already present at an early postnatal age. To gain knowledge about this developmental impairment, we defined lipid groups based on their trajectory pattern over time (Figure 1d,e and Appendix A). We identified two groups displaying a significant difference between WT and CMT1A myelin: group A, consisting of HexCers and long-chain SMs, whose concentration markedly increases in WT but not in CMT1A myelin; and group B, including a subgroup of phosphatidylcholines (PC) and phosphatidylethanolamines (PE), whose concentration decreases during development remaining however significantly higher in CMT1A (Figure 1d,e). We also identified two additional groups displaying a similar developmental trajectory in the two genotypes: group C, consisting of long-chain poly-unsaturated triglycerides (TG), whose concentration increases during development; and group D, including other subgroups of PC and PE, whose concentration decreases during development (Figure 1d,e).

To further investigate the biological meaning of these results, we performed a lipid ontology search on the four groups. Interestingly, group A was significantly enriched in saturated and mono-unsaturated very long-chain fatty acids (C22–24), whilst group B was enriched in short-chain fatty acids [23,24]. Finally, group C and group D were enriched in triacylglycerols and unsaturated short- and medium-chain fatty acids (C18 and C20), respectively (Figure 1d,e, and Appendix A).

Overall, lipidomic profiling revealed a delayed maturation of CMT1A myelin already detectable at P10 which becomes substantial at P20. In particular, CMT1A myelin is deprived of lipid species that are deemed important to guarantee optimal structural and electrophysiological properties, while displaying higher levels of lipids typical of unspecialized plasma membranes [23,25,26].

### 2.2. Maturation of Myelinated Fibers Structure Is Delayed during Development in CMT1A

We then paralleled lipidomic data with an extensive quantitative neuropathology study from P5 to P365, to pinpoint when, in time, specifically associated morphological and lipidomic alterations occur. Given that we identified P20 as a critical time point for the maturation of myelin lipid composition, we especially monitored the P10–P20 time frame and we added the P15 timepoint to the morphometric analysis performed over time. We analyzed nine different geometric parameters of over 130.000 fibers, using an ad hoc macro-based algorithm that we established in our laboratory, using the Image Pro-Plus Software version 7.0 (Immagini e Computer, Rho, Milan, Italy) (see Section 4 and Appendix A and Methods for details).

We found that, from P15–P20 onwards, CMT1A nerves display reduced axon and fiber diameter (Figure 2a,b and Appendix A). Moreover, the myelinated area is lower in CMT1A compared to WT (Figure 2a,b), while fiber density and roundness are not subjected to any relevant change (Appendix A). We also identified and quantified a subset of myelinating SCs that we called “hyperintense cells” due to their dark-blue appearance following toluidine blue staining (Appendix A). The number of these cells is high at immature stages of development (P5), and decreases over time albeit to a lesser extent in CMT1A than in WT nerves, especially at P10–P20 (Appendix A). Electron microscopy micrographs showed that “hyperintense cells” are characterized by an extensive endoplasmic reticulum network, which has been shown to be associated with high protein and lipid biosynthesis (Appendix A) [27,28].

Given that axonal diameter and myelin thickness increase during physiological development and deeply influence nerve conduction velocity, we examined these parameters in WT and mutant animals. First, to characterize axonal growth, we analyzed the axon diameter frequency distribution at each time point. The results show that in WT animals the frequency distribution steadily shifts towards larger caliber axons until P180, and then stabilizes (Figure 3A). Instead, in CMT1A animals the frequency distribution remains the same until P20, and then displays just a small shift between P30 and P365 (Figure 3A). Overall, these data indicate that axon growth in CMT1A animals is severely impaired and delayed compared to the WT animals; as a result, juvenile and adult CMT1A nerves show a progressive accrual of small-caliber fibers which account for more than half of all MFs, and a concomitant deprivation of large-caliber fibers (Figure 3B). As an example, at P180 and P365, the percentage of fibers with an axon diameter larger than 4 μm (that we assumed as a threshold to divide small- from medium/large-caliber axons) was above 80% in WT nerves, whilst below 60% in CMT1A nerves (Figure 3B).

Then, in order to monitor myelin development and characterize the growth of the myelin sheath, we analyzed the average g-ratio of WT and CMT1A fibers, following their grouping based on axonal diameter. This latter aspect is crucial for the correct description and interpretation of results; in fact, since g-ratio strongly depends on axon caliber during development (Appendix A), only g-ratios measured from axons of the same size and at the same developmental stage can be meaningfully compared. We found signs of hypo- (increased g-ratio) and hypermyelination (decreased g-ratio) in CMT1A fibers already at P10 (Figure 3C and Appendix A). In particular, the largest fibers are significantly reduced and hypomyelinated in CMT1A compared to WT at each time point, while the smallest ones are increased and hypermyelinated. Interestingly, small/medium-caliber fibers (2–5 μm of axon diameter), which are the largest fibers at early time points and the smallest at later stages, are hypomyelinated in young CMT1A nerves, but become hypermyelinated at later stages (Appendix A). This result suggests that myelination in CMT1A nerves progresses at a different pace than in the WT nerves and that it possibly also occurs in a shifted time window. All this evidence indicates that CMT1A axons show a delayed and defective growth; consequently, CMT1A nerves display an accumulation of small-caliber fibers, which are also hypermyelinated.

## 3. Discussion

In this study, we monitored the maturation of peripheral myelin lipid composition in WT and CMT1A rats and we highlighted an early developmental impairment of CMT1A myelin. Moreover, we analyzed the evolution of morphological parameters in MFs and consistently found that CMT1A axons and myelin sheaths also display defective growth which is evident soon after birth.

Myelination is a complex and finely regulated process that begins during the early stages of development in mammals and gives rise to long-lasting membrane structures [29,30,31]. In particular, the maturation of human PNS begins throughout early intrauterine life: NCV steadily increases between 24 and 40 weeks of gestation, reaching values of 20–25 m/s at birth [32], and approaching adult values (50–60 m/s) at 2–4 years of age, concomitantly with the anatomical maturation of peripheral nerves and the increased motor skills of the child [33,34]. In line with that, the structural maturation of peripheral nerves consists of a doubling in axonal diameter between 5 months and 4 years, an increase of myelin thickness in the first 4–5 years, and an elongation of internodes until the second decade of life by more than a factor of 4 [33,34,35].

CMT1A patients display a remarkable, uniform, and persisting NCV slowing (20 m/s) by the age of 2 years [3,5,6,7,8]; moreover, CMT1A sensory and motor nerves show structural abnormalities, including reduced myelin thickness, shortening of internodal length, loss of large-diameter fibers, and onion bulb appearances [8,36]. Interestingly, CMT1A children under 1 year old have normal or almost normal NCV and structural organization of peripheral nerves comparable to healthy subjects of equal age [36,37]. Therefore, NCV studies and neuropathological evidence support the hypothesis of defective PNS development in CMT1A likely involving both axon and myelin maturation.

To date, chemical features of myelin during PNS development have been poorly investigated as compared to the central nervous system (CNS) and the available studies mainly concern adult animals or aging processes analyzed by transcriptomic or proteomic approaches [38,39,40,41,42,43].

We analyzed the whole lipidome of peripheral myelin in WT and CMT1A rats during PNS development. This allowed us to characterize for the first time the evolution of lipid composition of WT peripheral myelin and to shed light on crucial aspects of CMT1A etiopathology.

We selected key time points in order to cover the optimal temporal window of PNS development. Indeed, at birth, the fibers of peripheral nerves are almost devoid of myelin and neurophysiological studies demonstrated that the NCV of the sciatic nerve in rats rapidly increases during the first three postnatal weeks, becoming 16 times faster than at birth and stabilizing later at P40–60 [44,45,46].

The results obtained from the lipidomic profiling show that PNS myelin in WT rats changes during development, shifting from a composition typical of unspecialized biological membranes to that of a specialized multilamellar myelin membrane. In particular, we observed an enrichment in very long-chain sphingolipids, as already demonstrated at the peak of myelination in the CNS of normal postnatal rats [38,47,48].

Transcriptomics data showed that CMT1A SCs fail to mount an adequate lipid biosynthetic program during development [21]. Our data corroborate this finding and are consistent with our previous results [22], highlighting that this impairment does not just rely on lipid shortage but rather on the delayed maturation of peripheral myelin composition in CMT1A. Although some alterations in CMT1A myelin are already present at P5, the difference between WT and CMT1A myelin lipid profiles becomes relevant between P10 and P20, concomitant with the myelination peak. This comes with no surprise, as defects in lipid metabolic pathways are exacerbated by the massive increase in lipid biosynthesis required at this stage [24].

It is also worth highlighting that, while WT myelin at P20 is quite different in lipid composition compared to earlier time points (i.e., P5 and P10), CMT1A myelin at P20 is still very similar to early postnatal myelin (Figure 1a). This suggests that CMT1A myelin maturation is impaired, and the time frame in which myelin maturation occurs might be shifted in CMT1A rats. From the molecular standpoint, CMT1A myelin displays an imbalance between long- and short/medium-chain lipids. Given that the high proportion of saturated long-chain lipid species likely confer to myelin their specific biophysical properties [23], including ion permeability, fluidity, and localization and maintenance of ion channel clusters [49,50,51], we speculate that this could be a critical aspect underlying the altered functionality of CMT1A myelin.

Taking into account the difference in developmental stages between rat and human species, the time interval P10–P20 in rats corresponds to an age of 1–2 years in humans [52,53]. Indeed, CMT1A children under 1 year old have a neurological picture comparable to healthy subjects of equal age; structural and neurophysiological abnormalities appear by the age of 2 years in these young patients, corresponding to the temporal window (P10–P20) in rats when the delay in myelin maturation is already appreciable. From a translational perspective, the consistency of our findings in rats with the progression of CMT1A in humans is quite remarkable.

Moreover, it is well known that the structural parameters of the fibers physiologically evolve during mammalian growth in order to allow for the efficient propagation of electrical impulses along the nerves [46,54]. In particular, axonal diameter is directly correlated to conduction velocity and, by axon–glia interaction, guides the radial and longitudinal growth of myelin, namely myelin thickness and internodal length, respectively [55,56,57]. In fact, as theoretically expected, among the different morphometric parameters analyzed, the most prominent defect we observed in CMT1A nerves was delayed and impaired axonal growth. Indeed, as we showed here, the lack of development of the largest MFs and the persistence of small fibers in adult age lead to a homogeneous cluster of immature fibers displaying electrophysiological properties (i.e., low conduction velocity) related to their small size in all peripheral nerves [46].

Furthermore, axonal defects have also been described in other murine models of peripheral neuropathies; for example, in mouse models of CMT1A, HNPP, and CMT4F the depletion of large-caliber axons and the shortening of internodal length are hallmarks that become evident with age and are present both in peripheral nerves and in ventral and dorsal roots [15,43]. Given that these different neuropathies share some neuropathological features, one could also hypothesize a common aberrant mechanism involving pathological axon–glia interactions and causing an immature-like state of the fiber bringing the attention to the axon as the primary target for therapeutic approaches.

In CMT1A nerves, small-caliber fibers also appear hypermyelinated and are mainly responsible for the decrease in the average g-ratio [9,10]. Whether hypermyelination originates from an increased number of stacked myelin lamellae or an altered compaction during myelination is still unknown. Certainly, this hypermyelination appears in a very early stage of development and persists in adult life without any positive effect on NCV.

To further support the immature phenotype of CMT1A nerves, we detected a prolonged settling of metabolically active Schwann cells (hyperintense cells), that progressively decrease in number during development and almost completely disappear after the first month of life, suggesting an important role just during maturation.

Our data also highlight a further evolution up to P180, indicating that PNS maturation is not complete after the myelination peak [58,59]. In this temporal window of six months, the rat completes skeletal muscle maturation and massively increases in body size and weight; realistically, the continuous evolution of PNS myelin lipid composition and structure that we observed parallel the physiological rat growth, as already described for the CNS [54,60]. This time interval is equivalent to adolescence/early adulthood in humans [53]. This observation is noteworthy because it offers a solid rationale for considering this extended temporal window effective for therapeutic intervention in CMT1A and other PNS myelin diseases.

## 4. Materials and Methods

### 4.1. Animal Model

The CMT rat, an animal model of CMT1A neuropathy originally developed in the K.-A. Nave laboratory, was used for the experiments. This transgenic rat has been generated by overexpressing the wild-type murine *Pmp22* gene in the rat genome [58]. Heterozygous CMT1A animals and wild-type (WT) littermates of both sexes at postnatal day (P) 5, P10, P15, P20, P30, P60, P75, P180, and P365 were used for the experiments.

### 4.2. PNS Myelin Isolation

Sciatic nerves from P5, P10, P20, P30, P60, and P180 rats were used to prepare myelin-enriched fractions as previously described [61,62]. We analyzed WT (n = 7) and CMT1A (n = 7) samples for each time point. To obtain a sufficient amount of purified myelin at P5, eight sciatic nerves from four different rats of the same sex and genotype were pulled together for each sample; for P10 and P20 myelin samples we used four nerves from two rats of the same sex and genotype; for P30 myelin samples we used two nerves from one rat; for P60 and P180 myelin samples, one nerve was used. Both male and female animals were included at each time point.

### 4.3. Lipid Extraction

Samples were added with 1 mL of IPA (previously added with 1 mL/mL of SPLASH^®^ Lipidomics^®^ Mass Spec Standard), incubated for 15 min on a shaker at room temperature, and then sonicated in ice-cold water for 10 min. After that, the suspension was centrifuged at 20,000× *g* for 20 min at 4 °C. The supernatant was then transferred into a new tube. The pellet was re-extracted with 100 mL of methanol: MTBE (1:1, *v*/*v*), incubated for 15 min on a shaker, and centrifuged at 20,000× *g* for 20 min at 4 °C. The supernatant was then pooled with the one collected in the previous extraction and dried under a nitrogen stream. Concentrated samples were resuspended with 50 μL of methanol: chloroform (9:1, *v*/*v*) for untargeted liquid chromatography with tandem mass spectrometry (LC-MS/MS) analyses. Samples were randomized and split into four different batches. Each batch included a consistent number of quality control (QC) samples (extracted from commercial mouse plasma), that were used to re-align and normalize data derived from different batches.

### 4.4. Untargeted LC-MS/MS Analyses

The lipidomic analyses were carried out on an ACQUITY UPLC system coupled to a Synapt G2 QToF high-resolution mass spectrometer (Waters, Milford, MA, USA), acquiring both in positive (ESI+) and negative (ESI−) ion modes.

Lipids were separated on a reversed-phase CSH C18 column (2.1 × 50 mm, 1.7 um), whose temperature was kept at 50 °C and eluted at a flow rate of 0.450 mL/min using the eluents: A = 10 mM ammonium formate in acetonitrile/water (60:40 *v*/*v*), B = 10 mM ammonium formate in isopropyl alcohol/acetonitrile (90:10 *v*/*v*). The gradient was the following: 0.0–1.0 min 10% B, 1.0–4.0 min 10 to 60% B, 4.0–8.0 min 60 to 75% B, 8.0–9.5 75 to 100% B and kept for 1.5 min. Then the column was reconditioned at 10% B for 2 min. The total run time was 13 min and the injection volume was set to 3 μL and 5 μL for acquisition in ESI+ and ESI−, respectively.

The scan range was set from 50–1200 m/z. Cone voltage was set at 35 V. Source temperature was set to 90 °C, desolvation temperature was set to 400 °C, and desolvation gas and cone gas (N_2_) flows were set to 600 and 30 L/h, respectively. The scan time was set to 0.3 s, low collision energy was set to 4 eV. Leucine enkephalin (2 ng/mL) was infused as lock mass for real-time spectra recalibration. MassLynx software version 4.1 (Waters) was used for data acquisition.

### 4.5. Data Processing

Features were extracted from raw data using TargetLynx software version 4.1 (Waters). Features with a signal-to-noise ratio below 10 were removed. Features with more than 50% of missing values and features with RSD (relative standard deviation) > 25% in the QC samples were removed. For the principal component analysis and heatmap visualization, data were log transformed and Pareto scaled.

### 4.6. Quantitative Neuropathology on Sciatic Nerves During Development

Morphological and morphometric studies were carried out on sciatic nerves from P5, P10, P15, P20, P30, P75, P180, and P365 WT (n = 3 for each time point) and CMT1A (n = 3 for each time point) rats fixed in 2.5% glutaraldehyde in cacodylate buffer and embedded in Epon. Semithin sections (0.8 μm) were cut, stained with toluidine blue, and photographed at a 1000× magnification with an Olympus PROVIS AX60 microscope connected to a JVC digital camera. Morphometric evaluation was performed using a custom macro based on the Image Pro-Plus Software version 7.0 (Immagini e Computer, Rho, Milan, Italy), which was able to extract nine different morphometric parameters from digitized images (see Appendix A). For each MF, the fiber perimeter and axon perimeters were traced, automatically obtaining the respective areas; fiber and axon diameters were derived from a circle of the corresponding area. The g-ratio (axon diameter/fiber diameter) was calculated and plotted as a function of axon diameter; fiber density and roundness (defined as π × diameter/perimeter) were also measured. Fibers associated with “hyperintense” cells (cells appearing dark blue following toluidine blue staining) were manually excluded, since they were not correctly segmented by our macro. Average values were obtained for all parameters for the two genotypes at each time point. At P5, at least 1000 MFs were evaluated for each rat; at later time points, at least 3000 MFs were evaluated for each rat, for a total of about 130,000 MFs.

### 4.7. Statistical Analysis and Data Visualization

To choose the appropriate statistical test, Shapiro–Wilk and Kolmogorov–Smirnov tests were used to test for normality, and an F test was used to test for equality of variance. For the comparison of morphometric parameters, one-way ANOVA was performed, followed by Sidak’s multiple comparison test; For the comparison of g-ratios, a Kruskal–Wallis test was performed, followed by Dunn’s multiple comparison test. The threshold for statistical significance was set at *p* < 0.05. For each experiment, the statistical test used is reported in the figure legends. Lipid enrichment analysis was performed using the LION web tool (LION/web|Lipid Ontology enrichment analysis) [63,64].

GraphPad Prism software version 9 (GraphPad Software Inc., San Diego, CA, USA) was used to perform all the statistical comparisons and to generate most of the graphs. The R packages factoextra (v. 1.0.7) (CRAN—Package factoextra (r-project.org) and ggplot2 (v. 3.4.4) (CRAN—Package ggplot2 (r-project.org) were used to generate the principal component analysis (PCA) plots of lipidomic data and the axon diameter frequency density plots. The heatmap of lipidomic data was generated using MetaboAnalyst (https://www.metaboanalyst.ca/MetaboAnalyst/, version 6.0). All figures were prepared using Adobe Illustrator 2023 (Adobe, San Jose, CA, USA).

## 5. Conclusions

Peripheral nerve development is a complex and dynamic process that is twisted at various levels in CMT1A neuropathy. In this study, we demonstrated a remarkable delay of myelin maturation and an arrest of axonal growth at an immature-like stage.

Our results corroborate the dysmyelinating phenotype for CMT1A and envisage CMT1A as a neurodevelopmental disease: in this inherited neuropathy, the PNS is arrested at early development stages and could be equated with that of a child. Overall, these aspects should be considered when evaluating new therapeutic options that have to be able to unlock this impasse by impacting the main populations of the PNS, i.e., neuronal axons and SCs, and their interaction.

## Figures and Tables

**Figure 1 ijms-25-11244-f001:**
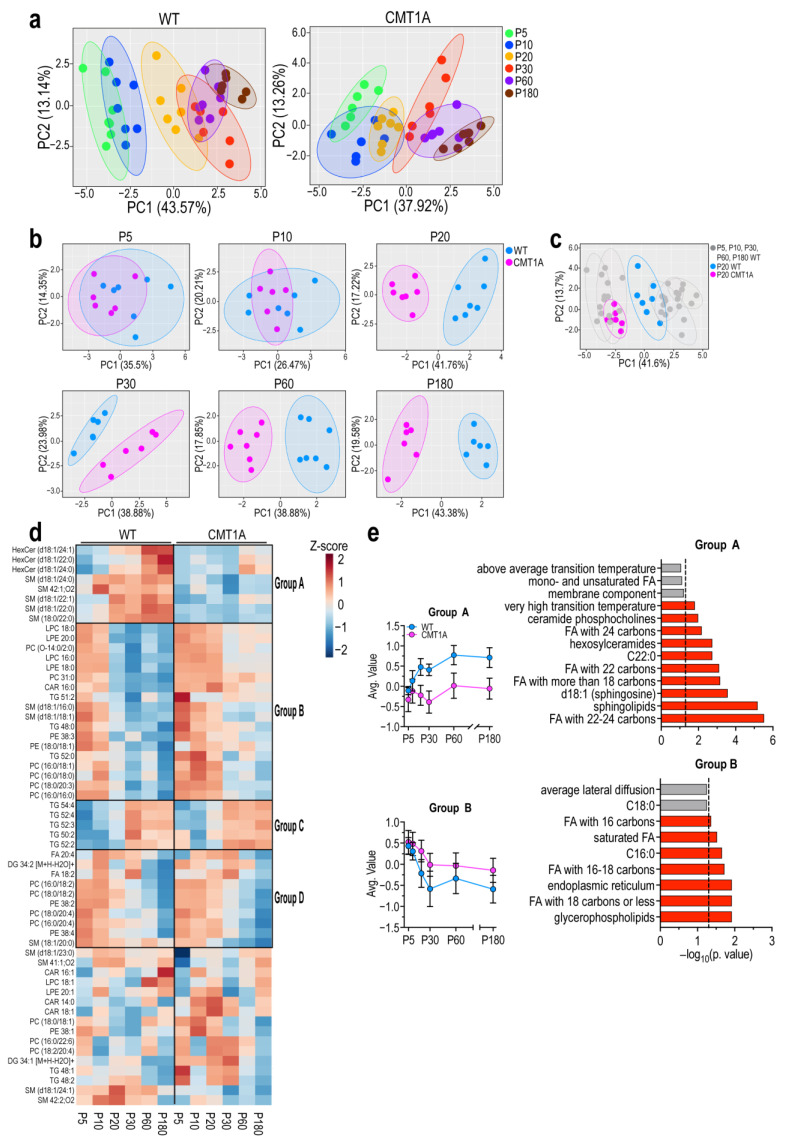
Impaired maturation of peripheral myelin lipid composition in CMT1A rats. (**a**) PCA plot of lipidomic data from WT (left) and CMT1A (right) purified peripheral myelin extracted from the sciatic nerve of P5, P10, P20, P30, P60, and P180 rats. n = at least 6 samples for each group; for each sample, several nerves from different rats were pulled together to obtain enough myelin. (**b**) Same data as in (**a**), but WT and CMT1A samples at the same time point are plotted together in a single PCA plot. (**c**) Same data as in A, but CMT1A P20 samples are plotted together with WT samples, to highlight the maturation delay experienced by CMT1A myelin at this time point. (**d**) Heatmap of lipidomic data. The four black rectangles highlight groups of lipids with a similar developmental trajectory (see also Appendix A for details). (**e**) On the left, average value of lipids in group A (top) and group B (bottom); on the right, results of enrichment analysis performed using LION software version 2023.04.14 on lipids in group A (top) and group B (bottom). The dotted line marks *p* value significance threshold (0.05).

**Figure 2 ijms-25-11244-f002:**
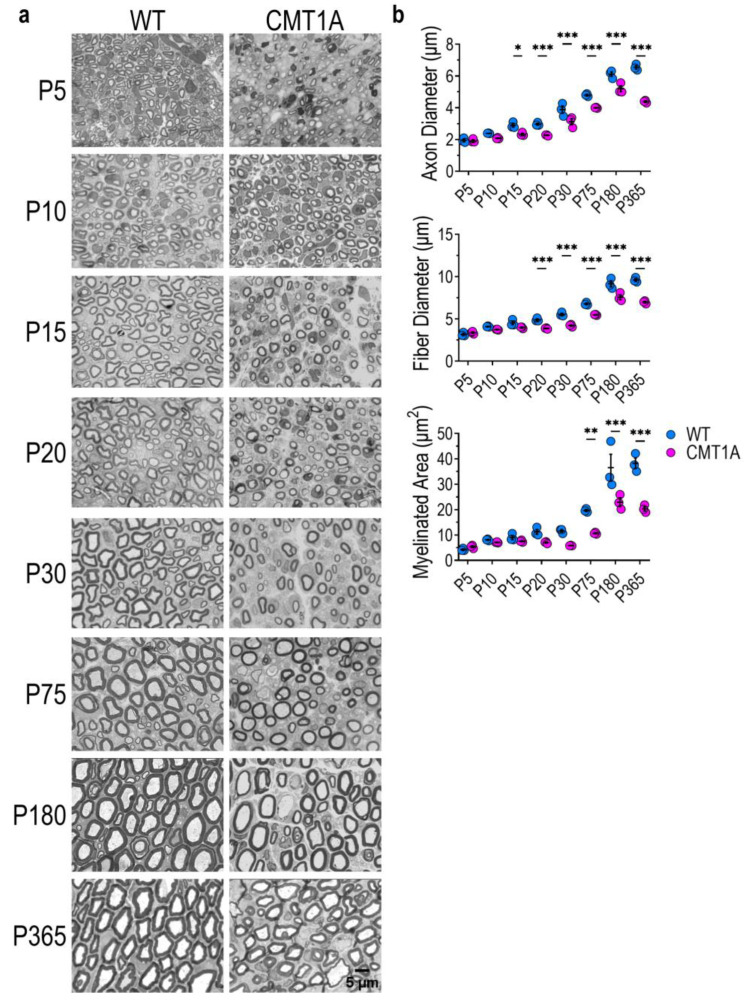
CMT1A myelinated fibers display impaired postnatal architectural development. (**a**) Micrographs of cross-sections of WT and CMT1A sciatic nerves at P5, P10, P15, P20, P30, P75, P180, and P365, stained with toluidine blue. Scale bar: 5 μm. (**b**) Morphometric parameters from WT and CMT1A sciatic nerve sections. n = 3 rats for each genotype and each time point. At P5, at least 800 myelinated fibers were evaluated for each rat; at later time points, at least 3000 fibers were evaluated for each rat. Values are presented as mean ± SEM. * *p* < 0.05, ** *p* < 0.01, *** *p* < 0.001. *p* values were calculated using one-way ANOVA followed by Sidak’s multiple comparisons test between the two genotypes at each time point.

**Figure 3 ijms-25-11244-f003:**
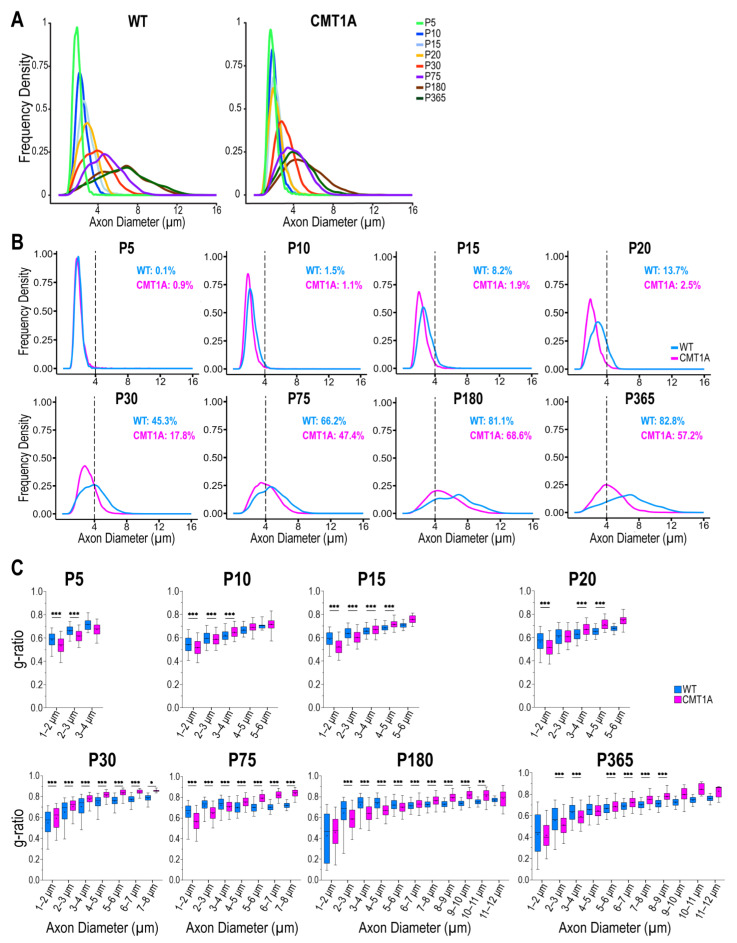
CMT1A myelinated fibers display impaired axon growth and aberrant myelination. (**A**) Frequency distribution of axon diameters in WT (left) and CMT1A (right) sciatic nerves at different developmental stages. (**B**) Same data as in (**A**), but the frequency distribution of WT and CMT1A axons at the same time point are plotted together. The dotted line marks the 4 μm threshold. The fraction of fibers with an axon diameter larger than 4 μm is reported in each graph. (**C**) Boxplots of the g-ratio for each fiber class in WT and CMT1A rats at different developmental stages. n = 3000; myelinated fibers from three different animals per genotype were evaluated at P5, 9000 myelinated fibers from three different animals per genotype were evaluated at later time points. The dot inside each box represents the mean. * *p* < 0.05, ** *p* < 0.01, *** *p* < 0.001. *p* values were calculated using the Kruskal–Wallis test followed by Sidak’s multiple comparisons test between the two genotypes for each axon diameter class.

## Data Availability

All the data supporting the findings of this study are available from the corresponding author upon request. Raw lipidomics data are included as Appendix A.

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
