# Peer review of "Monitoring Myelin Lipid Composition and the Structure of Myelinated Fibers Reveals a Maturation Delay in CMT1A"

_ijms, 2024, doi:10.3390/ijms252011244_

Round 1

Reviewer 1 Report

Comments and Suggestions for Authors

Examing a well established rat model of CMT1A, the authors examined myelin lipid changes at very early time points of postnatal development and could correlate changes in the myelin lipid maturation profile with impaired maturation at the ultrastructural level of myelin. Methods and Results are well described and the analytic methods completely appropriate. The authors provide strong evidence for a significantly delayed myelin maturation in the CMT1A rat model.

Comments:

Introduction:

It might be helpful for readers if the authors would briefly mention the genetic cause of CMT1A (PMP22 Gene) and the transgenic modifications in the rat model used in their study.

Results:

One relatively abundant group of lipids in myelin are sulfatides; the authors did not analyse this lipid: could the authors comment on this (is there a specific reason that this lipid was not measured?)

There should be a reference to the supplemental Table 2 in the legend to Figure 1d (or would it be possible to label the heatmap with the lipid names within the Figure 1d?).

One important question was not addressed by the authors: does the delay in maturation also correlate with altered PMP22 protein levels? Maybe PMP22 protein levels at the time points examined by the authors had already been published: if so, please give references for this; otherwise PMP22 levels could be examined by Western blot analysis. That would significantly improve the manuscript.

Reviewer 2 Report

Comments and Suggestions for Authors

In this research manuscript the authors demonstrated developmental
myelin dysfunction in a CMT1A disease model in rats. I believe the topic
addresses a specific gap in the field, even though it uses an animal
model. Introduction is clear and the aims are declared. Background
information on CMT1A was enough because the authors did use this model
to show myelin dysfunction in early development. Nevertheless, it would
be more interesting if the authors analyzed sulfatide lipids of myelin,
too. Figures 1-3 are well-presented and effectively represent this
study. Methods are descriptive and research plan is well designed and
performed. Discussion is robust. This research study is the natural
conclusion of a previous research already published by the authors. The
conclusions consistent with the presented evidence.

Round 2

Reviewer 1 Report

Comments and Suggestions for Authors

The essential points of the first review have been satisfactorily adressed by the authors. This reviewer suggests to accept the manuscript in the present form.